# Decadal Change in Seroprevalence of Chikungunya Virus Infection in Pune City, India

**DOI:** 10.3390/v14050998

**Published:** 2022-05-07

**Authors:** Shilpa Jagatram Tomar, Kalichamy Alagarasu, Ashwini More, Manasi Nadkarni, Rupali Bachal, Minal Bote, Jayashri Patil, Vasanthy Venkatesh, Deepti Parashar, Babasaheb Vishwanath Tandale

**Affiliations:** 1Epidemiology Group, ICMR-National Institute of Virology, Pune 411021, India; drshilpatomar.niv@gmail.com (S.J.T.); vasanthykmr@gmail.com (V.V.); 2Dengue and Chikungunya Group, ICMR-National Institute of Virology, Pune 411001, India; alagarasu@gmail.com (K.A.); ashwini05.s@gmail.com (A.M.); ebcmanasi@gmail.com (M.N.); rupaliniv@yahoo.co.in (R.B.); minalbote08@gmail.com (M.B.); jayashrinivnivniv@gmail.com (J.P.)

**Keywords:** seroprevalence, chikungunya virus, India, vector-borne diseases, *Aedes*

## Abstract

Chikungunya virus (CHIKV) is an arthropod-borne virus capable of causing large outbreaks. We aimed to determine the decadal change in the extent of chikungunya virus infection from 2009 to 2019. We implemented a prospective cross-sectional survey in Pune City using a 30-cluster approach with probability-proportion-to-size (PPS) sampling, with blood samples collected from 1654 participants in early 2019. The study also included an additional 799 blood samples from an earlier serosurvey in late 2009. The samples were tested by an in-house anti-CHIKV IgG ELISA assay. The overall seroprevalence in 2019 was 53.2% (95% CI 50.7–55.6) as against 8.5% (95% CI 6.5–10.4) in 2009. A fivefold increase in seroprevalence was observed in a decade (*p* < 0.00001). The seroprevalence increased significantly with age; however, it did not differ between genders. Modeling of age-stratified seroprevalence data from 2019 coincided with a recent outbreak in 2016 followed by the low-level circulation. The mean estimated force of infection during the outbreak was 35.8% (95% CI 2.9–41.2), and it was 1.2% after the outbreak. To conclude, the study reports a fivefold increase in the seroprevalence of chikungunya infection over a decade in Pune City. The modeling approach considering intermittent outbreaks with continuous low-level circulation was a better fit and coincided with a recent outbreak reported in 2016. Community engagement and effective vector control measures are needed to avert future chikungunya outbreaks.

## 1. Introduction

Chikungunya (CHIK) is a viral disease caused by the chikungunya virus (CHIKV), a single-stranded positive-sense enveloped RNA virus of the genus *Alphavirus*, family Togaviridae, and is transmitted by the bite of female *Aedes* mosquitoes. Globally, an estimated 1.3 billion people in 94 countries are at risk of chikungunya virus (CHIKV) infection [1]. In India, the first confirmed outbreak was reported from Calcutta in 1963, followed by epidemics in various states. From 1963 to 1973, the first wave of CHIKV outbreaks caused by the Asian lineage of CHIKV was reported from India [2]. After 1973, no major outbreak was reported for a span of 32 years.

The virus re-emerged in 2006 in Andhra Pradesh, which began near the coast of Kenya in 2004, spreading towards the Indian Ocean islands, affecting nearly 1.4 million people before spreading further [3,4,5]. Since 2006, CHIKV outbreaks have been reported from different parts of India, with several cases during 2016 [6]. A large outbreak of chikungunya was reported in 2016 in Pune, a metropolitan city in the state of Maharashtra, Western India [7]. Even during periods of silence, the virus is maintained by a low level of transmission in nature in an occult cycle, or by transovarial transmission (TOT) [8]. At present, chikungunya is endemic in 24 states of India and six union territories, highlighting it as an important public health problem. No licensed vaccine is available to date, although many potential candidates have been identified. In India, the research focus has additionally been on the traditional system of medicines, which has shown promising results [6].

Variation in the CHIKV genotype leading to improved transmission efficiency of the vector mosquitoes, urbanization, population growth and the build-up of vast numbers of immunologically naïve individuals in the community have been associated with the re-emergence of the virus [1,9]. In India, lower percentage positivity has been recorded during summer, increasing as monsoon sets in and continuing to be high in winter [6].

A serosurvey undertaken in Southern India provided insight into the more recent and epidemic transmission of chikungunya infection, in contrast to very high endemic dengue transmission, highlighting the need for enhanced surveillance to further implement control measures [10]. Periodic seroprevalence surveys are therefore important tools to supplement routine surveillance activities, providing insights into CHIKV circulation, immunity at the population level and the risk of future outbreaks.

In India, only a handful of CHIKV seroprevalence surveys have been done to date, usually limited to a geographical area or after a CHIKV epidemic [10,11,12,13]. There is a gap in the information domain about the seroprevalence of chikungunya in Western India, as the majority of the studies have been carried out in Southern India. Moreover, the studies carried out in the western region are hospital- or facility-based, with a focus on specific population groups, and are unable to provide the necessary information in a generalizable manner [11,14].

Therefore, a population-based house-to-house serosurvey study was carried out in Pune to bridge the aforementioned gap by generating seroprevalence data for chikungunya infections in Western India. This study complements the traditional approach of symptom- and laboratory-based surveillance and provides an indirect way of monitoring infections in the population. We also assessed the change in seroprevalence over 10 years in Pune using retrospective specimens.

## 2. Materials and Methods

### 2.1. Study Site

The study was carried out in Pune, the seventh-largest metropolis in India and the second largest in the state of Maharashtra. It has a reported population of 3,119,901 as of 2011 (Census 2011 of India). Pune City is placed under the administration of Pune Municipal Corporation (PMC) and is divided into 15 broader administrative zones, namely Aundh, Bhavani Peth, Kothrud, Warje, Hadapsar, Bibvewadi, Dhankawadi, Shahkarnagar, Kasba Vishram Baug, Tilak Road, Dhole Patil Road, Nagar Road, Gholeroad, Yerwada and Sangamwadi (Figure 1). These broader units are further divided into 144 wards for administrative purposes. Each ward was considered as an individual cluster and 30 wards were selected as the study area.

Study design and sample size: From the samples collected during an age-stratified serosurvey conducted in 2009 in Pune employing a 30-cluster methodology, a sample size of 799 subjects covering a proportionate number of different age groups (0–60) was estimated assuming 50% seroprevalence, 5% error and a design effect of 2 for cluster sampling. From the baseline serosurvey samples (*n* = 2520) collected from Pune City in October 2009 for H1N1 [15], 799 age-representative samples were tested for chikungunya antibodies in this study. The samples that were available in sufficient quantity were randomly selected, with appropriate representation of age groups and locality.

For the prospective survey conducted in Pune in early 2019, additional subjects were considered for making comparisons among different age groups and as per high, moderate and low disease incidence levels. The estimated target sample size was 1600 participants with proportionate allocation in different age groups. The sampling strategy employed in this study was randomized multistage cluster sampling, developed and elaborated by the Expanded Programme on Immunization (EPI) of the World Health Organization (WHO) [16]. This sampling technique involved the listing of the 144 wards of Pune City. Further, thirty clusters were selected from a sampling frame of 144 clusters (i.e., wards) by probability-proportional-to-size (PPS) sampling [16] (Figure 2). The PPS sampling was implemented and the locations were spatially distributed to have geographic representation. After listing the households, from each age group, a pre-determined number of samples were selected by systematic sampling, such that the samples in each age group reflected the age-wise stratification as observed in Pune City. An effort was made to survey most of the household members to have the age and gender representation of the population. This population-based house-to-house survey was conducted from 18 March 2019 to 10 April 2019, in which 1654 participants were recruited during the course of the study.

Sample collection: Samples were collected from the study population after administration of informed consent forms. Approximately 3–5 mL of blood was collected from adults and 2–3 mL from children (Figure 2). A maximum of two attempts were made to collect blood from the participants. Serum was separated by centrifuging at 2000 rpm for 2 min, aliquoted and stored at −80 °C in the deep freezer. The study was reviewed and approved by the Institutional Human Ethics Committee, ICMR—National Institute of Virology, Pune (NIV/IHEC/2018/Dec/D-6).

### 2.2. Detection of Chikungunya-Specific IgG Antibodies Using an In-House Anti-CHIKV IgG ELISA Assay

Testing was conducted at ICMR—National Institute of Virology, Pune using an in-house IgG ELISA assay that operates on the principle of direct ELISA, as described earlier [11]. Using a focus reduction neutralization assay as a reference, the assay had a sensitivity of 87.7% (95% CI 75.2–95.3) and a specificity of 88.6% (95% CI 78.7–94.9).

Briefly, semi-purified CHIK virus was coated in a 96-well micro-titer plate (Maxisorp, Nunc; Denmark) and wrapped in aluminum foil and incubated at 4 °C overnight. The coated wells were blocked with blocking solution at 37 °C for half an hour. After washing three times with wash buffer, 100 µL of 1:100 diluted serum samples, positive controls and negative controls were added to the microtiter plate and incubated at 37 °C for half an hour. Post incubation, the plate was washed five times with wash buffer. Horseradish peroxidase (HRP) conjugated goat anti-human IgG antibody (Sigma Chemicals, St. Louis, MO, USA) was then added and the plate was incubated at 37 °C for half an hour. Post incubation, the plate was washed and the substrate solution (TMB) was added to the plate. The plate was incubated in the dark for 5–10 min, the observed reaction was stopped by the addition of 1N H_2_SO_4_, and the absorbance was measured at 450 nm. The cut-off value for seropositivity was calculated by multiplying the mean optical density of negative controls by three. Samples that showed an optical density equal to or above the cut-off were considered positive, while samples with optical density values less than the cut-off were reported as negative.

### 2.3. Detection of Chikungunya IgM by Capture ELISA

To determine whether any subject had a serological response indicative of recent infection, the serum samples were tested for the presence of anti-CHIKV IgM antibodies using the CHIKjj Detect™ IgM ELISA Kit (Inbios International Inc., Seattle, WA, USA). The test was performed and interpreted according to the instructions provided by the manufacturer. As per the manufacturer’s claim, the kit has a sensitivity and specificity above 90%.

### 2.4. Estimation of Annual Force of Infection Using Age-Stratified Serological Data

Based on the 2019 data, annual force of infection was estimated using serocatalytic models employing the Rsero package [17]. Three models were considered for calculating the annual force of infection: constant model, outbreak model and combination of constant and outbreak model. In the constant model, the force of infection remained the same/similar through the years. In the outbreak model, a recent chikungunya outbreak leading to infection of all the subjects was assumed, while in the combination of the constant and outbreak model, a recent outbreak was followed by a constant force of infection. The models were compared using the deviation information criterion (DIC).

### 2.5. Statistical Analysis

The online software OpenEpi version 3.01 was used for the statistical analysis. Seroprevalence percentages among different age groups were determined with 95% confidence intervals. Comparison of seroprevalence in different age groups was done by testing proportions among age groups. A *p*-value of less than 0.05 was taken as significant. All the analyses related to calculating the force of infection were done using R version 3.6.1, employing the Rsero package [17].

## 3. Results

### 3.1. Seroprevalence among Study Participants

In 2009, the overall seroprevalence was 8.5% (95% CI 6.5–10.4). The seroprevalence among males (9.1%, 95% CI 6.0–12.1) and females (8.0%, 5.5–10.5) was similar. The age-specific seroprevalence in children aged 0–9 years was 7.3 % (95% CI 2.4–12.1). In children aged 10–19 years, seroprevalence was 4.2 % (95% CI 1.5–6.8). The highest seroprevalence of 12.1% (95% CI 7.0–17.0) was seen in participants in the age group of 20–29 years.

In 2019, the overall seroprevalence was 53.2% (95% CI 50.7–55.6). The seroprevalence among males (52.5%; 95% CI 48.9–56.2) was similar to that of females (53.7%; 95% CI 50.5–56.9). The age-specific seroprevalence in children aged 0–9 years was the lowest (37.8%; 95% CI 32–44). In children aged 10–19 years, seroprevalence was 46.9% (95% CI 42.5–51.3) (Table 1).

During the study, the clinical history of the participants was sought and they were asked if they had experienced any febrile illness in the past 3 months, to help document continuous or high infection transmission levels, in case they existed. Of the 1654 participants, only nine participants recalled having suffered from febrile illness, and the samples of these participants, along with their family members (*n* = 21), were tested for CHIKV IgM antibodies. Of these 21 participants, one of the asymptomatic family members was positive for CHIKV IgM antibodies reflecting recent or current infection.

The CHIKV seroprevalence in different administrative wards of Pune in 2019 varied from the lowest of 23.5% in Aundh to the highest in Dhole Patil at 87.5%. Out of the total 15 zones, eight had a seroprevalence of more than 50%. Of the 30 clusters, 15 were designated slum areas as per the Pune Municipal Corporation and the remaining were non-slum areas. The seroprevalence in slum areas was 65.2% (95% CI 61.7–68.6) vs. 56.7% (95% CI 53.1–60.4) in non-slum areas (*p* value = 0.00097; Odds ratio = 1.43, 95% CI 1.1–1.7). The seroprevalence was the highest in central zones, followed by northern, eastern and southern regions. The seroprevalence was the lowest in the northwestern zones, followed by the southwestern zones (Figure 3).

### 3.2. Comparisons of Seroprevalence: Pune 2009 and 2019

The overall seroprevalence of 53.2% (95% CI 50.7–55.6) in 2019 was significantly higher (*p* < 0.0001) as compared to 2009 (8.5%, 95% CI 6.5–10.4). This significant increase was seen both in males (52.5%; 95% CI 48.9–56.2 vs. 9.1%, 95% CI 6.0–12.1) and females (53.7%; 95% CI 50.5–56.9 vs. 8.0%, 5.5–10.5) in 2019 and 2009, respectively. Age-group-wise seroprevalence also revealed a significant increase in 2019 compared to 2009. All the age groups reported a fivefold increase in seroprevalence, while the increase was 11-fold in the 10–19-year-old group (Table 1).

### 3.3. Annual Force of Infection

The annual force of infection was estimated using three serocatalytic models based on the serological data for the year 2019. The annual probability of infection under the constant model with 95% CI was 3.05 (2.85–3.27) (Figure 4A). Under the outbreak model, the outbreak probability of infection with 95% CI was 53.88 (51.40–56.26) (Figure 4B). The mean outbreak was predicted as 2014. In the combination of the constant and outbreak model, the mean annual probability of infection during the outbreak was 35.85 (2.95–41.19) and the annual probability of infection thereafter was 1.27 (0.019–1.63) (Figure 4C). The mean outbreak year was predicted as 2016. The combination of the constant and outbreak model had the lowest DIC value and was considered as the best fit.

The incidence of chikungunya cases reported by the Pune Municipal Corporation for years 2012–2019 revealed an outbreak in 2016 (http://opendata.punecorporation.org/Citizen/CitizenDatasets/Index) (accessed on 27 August 2021) (Figure 4D), also coinciding with the prediction of the combination of the outbreak and constant model.

## 4. Discussion

Seroprevalence studies help in assessing the extent of infection by a particular pathogen within the community so as to ensure the implementation of appropriate measures to prevent outbreaks. Moreover, seroprevalence studies help to understand the history of the circulation of the pathogen [18]. In the present study, a prospective cross-sectional study in Pune City using a 30-cluster sampling approach with probability-proportion-to-size (PPS) sampling was carried out during early 2019 to determine the seroprevalence of chikungunya in Pune, the second-largest metropolis in Maharashtra. This cluster-sample design is the most practical solution for most surveys, where taking a simple random sample from individuals across the country would be practically impossible. It has been extended to many health surveys, with different aims, owing to its simplistic appeal. In this technique, communities are selected with probability proportional to size, with each household having an equal probability of being selected, being a “self-weighting” procedure [16]. This sampling technique may provide less precision in comparison to simple random sampling or stratified sampling, but this loss in precision is offset when an appropriately large sample size is selected.

Moreover, samples collected from Pune City in October 2009 for the H1N1 serosurvey [15] were also investigated to assess the change in the seroprevalence of chikungunya over a decade. The results revealed an overall seroprevalence of 8.5% in 2009 and 53.2% in 2019, suggesting a fivefold increase in chikungunya seroprevalence over this decade. A retrospective study carried out using samples collected during 2017 in Pune has reported an overall seroprevalence of 45.7%, while another nationwide survey conducted in 2017 has reported a seroprevalence of 37% [11,12]. Factors such as urbanization, population growth and climatic conditions favor vector breeding, and a large proportion of the susceptible population, as evidenced by the low seroprevalence during 2009, might have contributed to the active circulation of chikungunya in the last decade in the study region. The seroprevalence increased with age in all the studies, including the present study, was lowest in children below nine years, and reflects the cumulative past exposure to CHIKV. In addition, adults, due to more movement, especially in different geographical areas or cities, are more likely to acquire infection in comparison to children, who are usually restricted to their households or their surrounding localities only.

Slum areas in Pune City showed a significantly higher seroprevalence of CHIKV as compared to non-slum areas, i.e., 65.2% (95% CI 61.7–68.6) vs. 56.7% (95% CI 53.1–60.4) in non-slum areas (*p* value = 0.00097, odds ratio = 1.43). Aundh, having the lowest seroprevalence (23.5%), has 22.05% of its population living in slums, with a slum density of 1171 persons/sq km. This is in comparison to Dhole Patil, having the highest seroprevalence (87.5%), where almost half the population, i.e., 49.97%, resides in slums, with a high slum density of 4545 persons/sq km [19]. This reflects the impact of impoverished living conditions and crowding on public health, highlighting the need to address poor living conditions and promote healthy and affordable housing, along with efforts for community engagement [20]. One of the limitations of our study was that we did not enquire regarding the distance of the workplace from the households and the mode of travel, which could have helped in providing information on population movement and its impact on disease transmission.

Modeling the circulation based on the seroprevalence data suggested the occurrence of a large outbreak of chikungunya during 2016, during which the annual force of infection was higher (~35.8% susceptible individuals seroconverted), followed by the low-level circulation of the pathogen in the following years (~1.2% of susceptible individuals seroconverted). To add support to the modeling results, the data from the National Vector-Borne Disease control programme for the years 2009–2018 indicate that a large number of chikungunya cases were reported during 2016 compared to previous years, suggesting a chikungunya epidemic in India [6]. Furthermore, data from our lab as well as incidence data from the Pune Municipal Corporation also indicated a chikungunya outbreak during 2016 in Pune [7,21]. In Western India, data from a national serosurvey conducted during 2017 indicated an annual force of infection of 1.7% using a constant model and an annual force of infection ranging between 0.5% and 4.3% for 1977–2017 using the heterogeneous model. The main limitations of the serocatalytic model used to predict the annual probability of infections is that these models are based on the assumption of long-lasting immunity and do not take into account either the waning antibody response in low-prevalence settings or the boosting of the antibody response due to repeated exposures in endemic settings. Moreover, the data are compressed into seronegative and seropositive states and the serocatalytic models are not effective in settings of high and moderate transmission intensity due to the long half-life of seropositivity and the plateau of high seroprevalence reached among young children [22]. The national serosurvey included three states from Western India, and the seroprevalence from Maharashtra was higher and was similar to that reported in the present study, except for the 0–9 years age group. The seroprevalence in the 0–9 years age group was 38% in the present study, while it was 10% in the national serosurvey. The national serosurvey included both urban and rural regions, as well multiple locations within the state, which might explain the lower seroprevalence in the lower age group [12]. It is possible that when a larger proportion of the susceptible population is built up within the community, epidemics might occur, followed by years of low-level circulation of the virus. Periodic serosurveys might help in identifying regions with lower seroprevalence, where the risk of future outbreaks might be greater, and enhancing the surveillance and vector control measures will help in averting large-scale outbreaks in such regions.

## 5. Conclusions

To conclude, the present study reports a fivefold increase in chikungunya infections over a decade in Pune, a metropolis in Western India. The study underscores the importance of periodical serosurveys to identify the regions at risk for future outbreaks to implement preventive measures, including creating awareness about chikungunya, its symptoms and its mode of transmission.

## Figures and Tables

**Figure 1 viruses-14-00998-f001:**
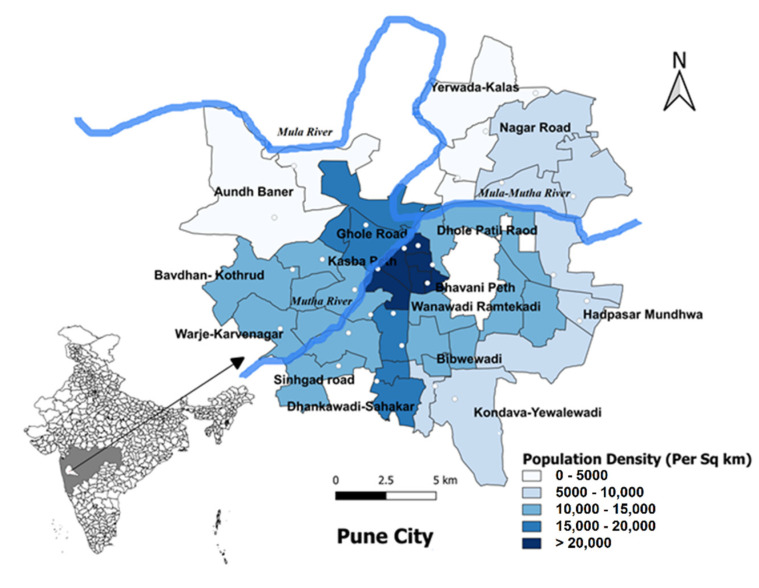
Georeference map of locations of 30 clusters surveyed in 15 administrative zones in Pune City, 2019. The administrative zones (*n* = 15) are highlighted in different shades of blue with the actual georeferenced locations of 30 clusters selected by using the probability-proportional-to-size (PPS) sampling. They are located on the map of Pune Municipal Corporation (PMC), Pune City for the second survey undertaken in 2019. The map also shows the Mula-Mutha river flowing through Pune City.

**Figure 2 viruses-14-00998-f002:**
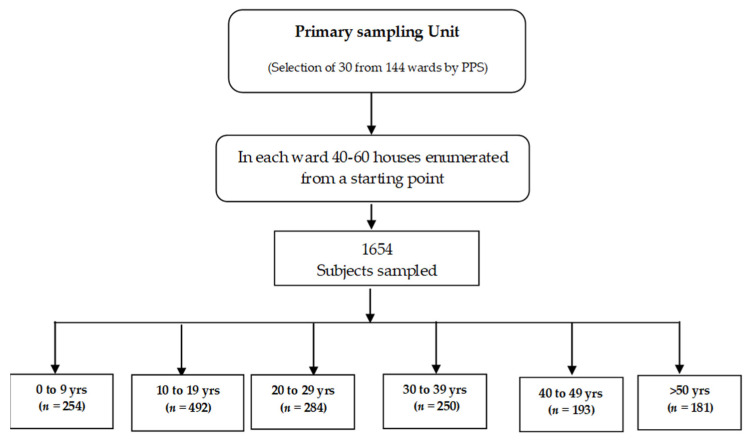
Flowchart for sampling design and profile of study participants.

**Figure 3 viruses-14-00998-f003:**
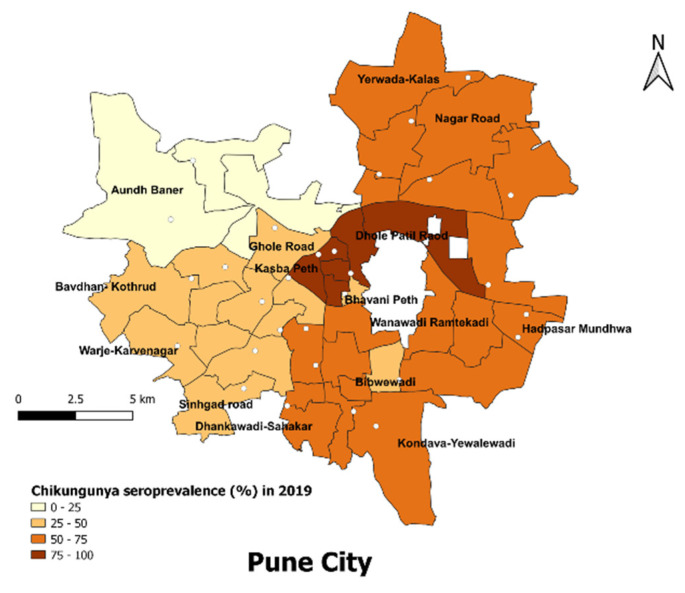
Seroprevalence (%) of chikungunya virus infection in 15 administrative zones of Pune City, 2019. The seroprevalence (%) of chikungunya virus infection is categorized and presented as different shades of orange; the highest seroprevalence is represented as dark brown and lighter shades depict lower seroprevalence levels. The vertical arrow indicates the north direction. The dots indicate the 30 cluster locations of the study.

**Figure 4 viruses-14-00998-f004:**
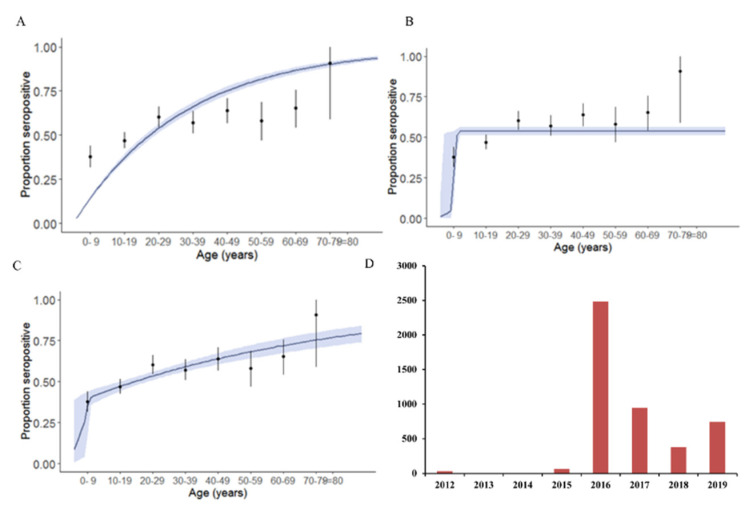
Seroprevalence of chikungunya predicted under different models: (**A**) constant model; (**B**) outbreak model; (**C**) combination of constant and outbreak model. Black line within the envelope (blue shade) represents force of infection and the envelope represents the 95% confidence intervals. Vertical line with black dots represents the observed seroprevalence with 95% CI. (**D**) represents the cumulative number of cases reported to Pune Municipal Corporation during the years 2012–2019.

**Table 1 viruses-14-00998-t001:** Comparison of age-stratified seroprevalence of CHIKV infection in Pune City in 2009 and 2019.

Age Group (Years)	2009		2019 *		Percent Increase	Fold Increase
	*n*	Anti-CHIK IgG (%)	*n*	Anti-CHIK IgG (%)		
0–9	110	8 (7.3)	254	96 (37.8)	30.5	5.2
10–19	214	9 (4.2)	492	231 (46.9)	42.7	11.1
20–29	149	18 (12.1)	284	171 (60.2)	48.1	5
30–39	131	13 (9.9)	250	143 (57.2)	47.3	5.7
40–49	80	7 (8.9)	193	124 (64.2)	55.3	7.2
≥50	115	13 (11.3)	181	115 (63.5)	52.2	5.61
Total	799	68 (8.5)	1654	880 (53.2)	44.7	6.25

* In 2019, children with age ≥5 years were sampled.

## Data Availability

Not applicable.

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
