# Peer review of "Decadal Change in Seroprevalence of Chikungunya Virus Infection in Pune City, India"

_viruses, 2022, doi:10.3390/v14050998_

Round 1

Reviewer 1 Report

In this article SJ Tomar et all compare the CHIKV seroprevalence between 2009 and 2019 in Pune city south west of India.

The presention of the objective and strategy used is clear but some assessement seems wrong, specifically page 2 line 53 in the introduction:

This is wrong to write following the paper of Rodrigues-Barraquer about the prevalence in Chennai that there is a massive burden of asymptomatic infection for CHIKV. This is thrue in this paper for Dengue (1% of past ilness reported) but not for CHIKV.

The sentence must be balenced as in the cited study in Chennai, the non-reported infection was indicated to be around 56%. This is not a massive burden taking in account that the study was performed in 2011-2012 thus 5 to 6 years after the epidemic. IN addition, many illness might be confonding. This is far from the data in the very same study about Dengue in which less than 1% of past illness were reported in the seropositive patient.

You must write «  up to 50% of infections are not remembered by patient 4 years latter by KAP survey.

The results are interesting but the presentation is of poor quality or more exactly impaired by the usage of different numbers in figures and presentation of the figures, In many cases, the reported number in the text or range are not the same.

To be clear in figure 1: They were 13 or 14 locations named in the map and not 15, the variable gray value do not correspond to the distinct ward but as shown in the figure 4 to the observed seroprevalence of CHIKV+

Some area are in white and to allow the reader to replace the Puna City geography you should add the location and figure of the Mutha river as well as a scale of distance on the left and not below. In this figure 1 you should indicate the population density or average of green area to full city in place of the gray scale that are also used in the very same figure 4 with a different legend.

For non indian people, a figure like the one i fund on the net and attached might help a lott to understand were is Puna City (at 600 M high).

http://image.slidesharecdn.com/na-oss2kunalkumar-29062015-150701090429-lva1-app6892/95/punes-trash-solution-a-zero-waste-city-by-kunal-kumar-municipal-commissioner-pune-municipal-corporation-3-638.jpg?cb=1435741584

IN material and method you must have to explain (or in the discussion in minima) the advance and limitation of the sampling method you use and provide a reference to a statistical epidemic method book as in page 3, lines 100-102

« randomised multistage cluster sampling. This sampling technique involved the 99 listing of the 144 wards of Pune City. Further… »

Finally you sampled a low number of ward considering the total this have to be commented.

IN result page 5 line 173, 175: I do not understand the meaning to speack of children below 9 years of age, between 9-17 than the children aged <18 years. This is confusing. You have to comment the figure with the numbers that are related to the age group you shown and not numbers that are not detailled in the figure 3.

About figure 4 you have to provide information correct and different than in figure 1 and not only the very same picture with a different legend. Scale continue to be important and following the various size and geometric forms of each tested ward a gradiant should be provided and not to color the whole ward with a single color when the selected point of sampling in in a corner. As it is performed in part in the paper from Rodriguez-Barraquer you cited as ref 10 in your introduction.

The comparison of prevalence between the age group is confusion in the sentence line 203-204: Please correct to be in agreement with the figure 4. In this figure the evolution is the opposite of what is described line 203 to 204.

Same in figure 6 ! It is not clear if the range 0-9 is really 0 to 9 or 5 to 9 years of age as indicated in the discussion. PLease explain and correct.

About the attitude towards Chikungunya in Pune City.

Page 246: Please conclude if it is a risk factor or not ! Are drum barrels open or closed ?

In the discussion a bit more carefull and please you should discuss the advantage and limitation of such a study if you do not performe the very same analysis during the two time points of the study. (line 252).

Please use the same age group in the figure, results text presentation and discussion

Minor comment:

Page1 line 35 : discard or rewrite the sentence that end with  “, and.”

Page 2 line 73 : what is “an estimated population of 31,19,901 as” ?

Page 5 line 151: If no data of KAP about Dengue there is no need to indictate this in the M&M

Page 7 line 199 there was a repetition “in 2009”.

Page 9 line 225 : please correct “in sync with “

Page 9 line 227: As noted in the M&M no comment about Dengue.

Page 246: Please conclude if it is a risk factor or not ! Are drum barrels open or closed ?

Page 276: complete “to years”.

Page 284: have check to the age group one more time.

Author Response

Comment 1: In this article SJ Tomar et all compare the CHIKV seroprevalence between 2009 and 2019 in Pune city south west of India.

The presentation of the objective and strategy used is clear but some assessment seems wrong, specifically page 2 line 53 in the introduction: This is wrong to write following the paper of Rodrigues-Barraquer about the prevalence in Chennai that there is a massive burden of asymptomatic infection for CHIKV. This is true in this paper for Dengue (1% of past illness reported) but not for CHIKV. The sentence must be balanced as in the cited study in Chennai, the non-reported infection was indicated to be around 56%. This is not a massive burden taking in account that the study was performed in 2011-2012 thus 5 to 6 years after the epidemic. In addition, many illness might be confounding. This is far from the data in the very same study about Dengue in which less than 1% of past illness were reported in the seropositive patient.

Response: We thank the reviewer for indicating our mistake. We have rectified this in the manuscript and added that in southern India, chikungunya has a more recent and epidemic transmission pattern in contrast to very high endemic dengue transmission. The text containing ‘massive asymptomatic chikungunya transmission’ has been deleted (Page 2, Lines 60-63).

Comment 2: You must write «  up to 50% of infections are not remembered by patient 4 years later by KAP survey.

Response: As suggested in the fourth report by reviewers, the KAP survey findings have been removed from the revised manuscript.

Comment 3: The results are interesting but the presentation is of poor quality or more exactly impaired by the usage of different numbers in figures and presentation of the figures, In many cases, the reported number in the text or range are not the same.

Response: Thank you for highlighting our mistake. We have now updated all the relevant figures and text to maintain uniformity throughout the manuscript.

Comment 4: To be clear in figure 1: They were 13 or 14 locations named in the map and not 15, the variable gray value do not correspond to the distinct ward but as shown in the figure 4 to the observed seroprevalence of CHIKV+

Response: Figure 1 has now been updated, having all 15 administrative zones; distinctly separated using different shades of blue, instead of the previously used grey gradient, as suggested by the reviewer (Page 3).

Comment 5: Some area are in white and to allow the reader to replace the Puna City geography you should add the location and figure of the Mutha river as well as a scale of distance on the left and not below. In this figure 1 you should indicate the population density or average of green area to full city in place of the grey scale that are also used in the very same figure 4 with a different legend. For non indian people, a figure like the one i found on the net and attached might help a lot to understand where is Puna City (at 600 M high).

http://image.slidesharecdn.com/na-oss2kunalkumar-29062015-150701090429-lva1-app6892/95/punes-trash-solution-a-zero-waste-city-by-kunal-kumar-municipal-commissioner-pune-municipal-corporation-3-638.jpg?cb=1435741584

Response: We have now modified Figure 1, including location of Pune City with reference to the Indian map, for better understanding. The population density of the 15 administrative zones of Pune city have also been added, as suggested (Page 3).

Comment 6: In material and method you must have to explain (or in the discussion in minima) the advance and limitation of the sampling method you use and provide a reference to a statistical epidemic method book as in page 3, lines 100-102

« randomised multistage cluster sampling. This sampling technique involved the 99 listing of the 144 wards of Pune City. Further… »Finally you sampled a low number of ward considering the total this have to be commented.

Response: As suggested, more details on the 30-cluster sampling approach with probability-proportion-to-size (PPS) sampling has been added in both methods (Page 4, Lines 111-112) and discussion. The advantages’ and limitations of this sampling method have been elaborated in the discussion (Page 11 Line 302-310). Relevant reference for the same have also been added (Page 14, Reference no. 16).

Comment 7: IN result page 5 line 173, 175: I do not understand the meaning to speak of children below 9 years of age, between 9-17 than the children aged <18 years. This is confusing. You have to comment the figure with the numbers that are related to the age group you shown and not numbers that are not detailed in the figure 3.

Response: Thank you for the suggestion. We have now re-categorised all age groups and maintained same groups throughout all figures (including modelling) and text.

As suggested in the fourth report, now both Figure 3 and 5 have been combined together and the data depicted in Table format, for clarity and better understanding (Page 6).

Comment 8: About figure 4 you have to provide information correct and different than in figure 1 and not only the very same picture with a different legend. Scale continue to be important and following the various size and geometric forms of each tested ward a gradiant should be provided and not to color the whole ward with a single color when the selected point of sampling in in a corner. As it is performed in part in the paper from Rodriguez-Barraquer you cited as ref 10 in your introduction.

Response: As suggested, Figure 1 (Page 3) and Figure 4 (Page 8) have now been modified incorporating inputs of the reviewers. In Figure 4, now gradient of orange colour has been used to depict seroprevalence in the 15 administrative zones of Pune city.

Comment 9: The comparison of prevalence between the age group is confusion in the sentence line 203-204: Please correct to be in agreement with the figure 4. In this figure the evolution is the opposite of what is described line 203 to 204.

Response: As suggested, we have now removed Figure 3 and 5and data has been combined in a table format for clarity, where the age groups have been re-categorised for uniformity and results updated (Page 6).

Comment 10: Same in figure 6 ! It is not clear if the range 0-9 is really 0 to 9 or 5 to 9 years of age as indicated in the discussion. PLease explain and correct.

Response: We had initially stratified our study age groups in accordance with a national serosurvey, for comparison, and presented those results in figures and text. But considering the age groups taken in our study for different prediction models, we have now taken the same age groups with a 10 year interval (0-9, 10-19 and so on) to maintain consistency.

Comment 11: About the attitude towards Chikungunya in Pune City.

Response: As suggested, we have dropped this component from the revised manuscript

Page 246: Please conclude if it is a risk factor or not ! Are drum barrels open or closed ?

Response: We have now removed KAP findings from our manuscript, as suggested.

Comment 12: In the discussion a bit more careful and please you should discuss the advantage and limitation of such a study if you do not perform the very same analysis during the two time points of the study. (line 252).

Response: As suggested, we have now added advantages and limitations of our study in the discussion (Page 11, Lines 302-310; Page 12, Lines 348-355). We would like to mention that in both time-points of the study, the sample collection was community-based and tested with the same strategy.

Comment 13: Please use the same age group in the figure, results text presentation and discussion

Response: We have now used the same age groups throughout the manuscript and also re-analysed the results accordingly, as indicated (Page 6, Table 1).

Minor comment:

Page1 line 35 : discard or rewrite the sentence that end with  “, and.”

Response: Discarded additional “and” (Page 1, Line 37)

Page 2 line 73 : what is “an estimated population of 31,19,901 as” ?

Response: Corrected to “reported” population (Page 2, Line 83)

Page 5 line 151: If no data of KAP about Dengue there is no need to indictate this in the M&M

Response: KAP findings removed from manuscript.

Page 7 line 199 there was a repetition “in 2009”.

Response: Correction made (Page 9, Line 246)

Page 9 line 225: please correct “in sync with “

Response: Correction made and modified to “coinciding with” (Page 10, Line 272)

Page 9 line 227: As noted in the M&M no comment about Dengue.

Response: KAP findings removed from manuscript.

Page 246: Please conclude if it is a risk factor or not ! Are drum barrels open or closed ?

Response: KAP findings removed from manuscript.

Page 276: complete “to years”.

Response: Corrected to “to previous years” (Page 11, Line 342)

Page 284: have check to the age group one more time.

Response: All age groups checked and updated in the revised manuscript.

Reviewer 2 Report

This study was conducted to examine the seroprevalence of chikungunya virus infection in India. More information and some correction are needed for this manuscript.

Introduction: More information about the epidemiology and current situation of CHIKV infection in India is needed.

Materials and methods: What are the sensitivity and specificity of ELISA for IgM and IgG used in this study.

Discussion: Please discuss the different sample sizes of age groups and the result of seroprevalence.

Others: Please check the consistency of using Chikungunya and chikungunya.

Line 34-35: Italic letter for genus name: Alphavirus, Aedes.

Line 131: 450 nm.

Author Response

This study was conducted to examine the seroprevalence of chikungunya virus infection in India. More information and some correction are needed for this manuscript.

Comment 1: Introduction: More information about the epidemiology and current situation of CHIKV infection in India is needed.

Response: As suggested, more information of CHIKV epidemiology and current situation in India added (Page 2, Lines 50-54, Lines 58-59).

Comment 2: Materials and methods: What are the sensitivity and specificity of ELISA for IgM and IgG used in this study.

Response: The sensitivity and specificity of ELISA for IgM and IgG kits used in this study has been added (Page 5, Lines 149-150).

Comment 3: Discussion: Please discuss the different sample sizes of age groups and the result of seroprevalence.

Response: All different sample sizes of age groups and results have now been compiled in Table 1 and elaborated in results (Page 6, Lines 200-203; 206-210).

Others: Please check the consistency of using Chikungunya and chikungunya.

Response: Same format for ‘chikungunya’ has now been used consistently throughout the manuscript.

Line 34-35: Italic letter for genus name: Alphavirus, Aedes.

Response: Letters italicized (Page 1, Line 36-37)

Line 131: 450 nm.

Response: Space added in 450 nm (Page 5, Line 161).

Reviewer 3 Report

A brief summary

The present study is a prospective cross-sectional study in Pune city (India) of Chikungunya virus using a model of seroprevalence with probability-proportion-to-size sampling during early 2019 (53.2%). Moreover, they also investigated the seroprevalence in 2009 (8.5%) suggesting a 5-fold increase in the seroprevalence over a decade. Testing was performed using in-house IgG ELISA assay and detection of chikungunya IgM ELISA (recent infection, only a few cases). The authors concluded that a larger % of naïve population within the community could lead to one epidemic after years of low-level circulation of virus and periodical serosurveys are needed.

The manuscript is extensive and the conclusions are moderate and appropriate. On the other hand, the authors do not explain the limitations of these types of predicted models.

No tables in this article.

Figures: in the figure 6, edit 6D. The figures in the manuscript are appropiate.

Author Response

A brief summary: The present study is a prospective cross-sectional study in Pune city (India) of Chikungunya virus using a model of seroprevalence with probability-proportion-to-size sampling during early 2019 (53.2%). Moreover, they also investigated the seroprevalence in 2009 (8.5%) suggesting a 5-fold increase in the seroprevalence over a decade. Testing was performed using in-house IgG ELISA assay and detection of chikungunya IgM ELISA (recent infection, only a few cases). The authors concluded that a larger % of naïve population within the community could lead to one epidemic after years of low-level circulation of virus and periodical serosurveys are needed.

Comment 1: The manuscript is extensive and the conclusions are moderate and appropriate. On the other hand, the authors do not explain the limitations o these types of predicted models.

Response: The limitations of predictive models has now been added to the manuscript, as suggested (Page 12, Lines 348-355).

Comment 2: No tables in this article.

Response: As suggested, Figure 3 and 5 have now been combined in a Table format (Page 6, Table 1).

Figures: in the figure 6, edit 6D. The figures in the manuscript are appropriate.

Response: Correction made in Figure 6 (Page 10).

Reviewer 4 Report

The study by Tomar et al. entitled “Decadal Change in Seroprevalence of Chikungunya Virus Infection in Pune City, India” provides information of CHIKV infection prior to and after an epidemic in Pune, India, 2016. While this study provides some additional information not previously published, the work adds only incrementally to a previous serosurvey done in 2017 by Patil et al. (Eur J Clin Microbiol Infect Dis. 2020 Oct;39(10):1925-1932) that show largely similar levels of past CHIKV infection. Further decreasing this reviewer’s enthusiasm, the inclusion of the KAP does not integrate
well and reads as an afterthought and brings little extra value to the manuscript as it is presented.

The manuscript would benefit greatly by condensing figures and reworking to provide a clearer focus.

General Concerns:

1) Having the 2009 data provides an interesting perspective regarding a potential level of immunity at which CHIKV is able to cause an epidemic (2016) but this is not developed in the manuscript. It is also unclear if the 2009 samples are similar to the 2019 samples. That is, are the 2009 samples from the community like the 2019 samples?

2) Figures 1 and 4 are nearly the same figure. It would be better to show the graph in figure 4 only.

3) Figures 3 and 5 would be more useful combined and in a table format. For Figure 5, a line graph is not appropriate as it demonstrates a linear increase from 2009 to 2019 which is not expected to be the case with an epidemic in 2016 as shown from data in Figure 6D.

4) It’s not clear how the KAP data integrates with the rest of the manuscript. It should either be better integrated somehow or reworked as a separate submission.

5) There is no discussion why certain wards show higher seroprevalence. For example, are these more impoverished wards, higher population density, etc? How does this translate into useful public health information going forward?

6) Why was an assessment of febrile illness in the past 3 months included in the manuscript when the primary objective is determining past infection (seroprevalence)? Inclusion of this detracts from the overall aim of the manuscript.

Author Response

The study by Tomar et al. entitled “Decadal Change in Seroprevalence of Chikungunya Virus Infection in Pune City, India” provides information of CHIKV infection prior to and after an epidemic in Pune, India, 2016. While this study provides some additional information not previously published, the work adds only incrementally to a previous serosurvey done in 2017 by Patil et al. (Eur J Clin Microbiol Infect Dis. 2020 Oct;39(10):1925-1932) that show largely similar levels of past CHIKV infection. Further decreasing this reviewer’s enthusiasm, the inclusion of the KAP does not integrate well and reads as an afterthought and brings little extra value to the manuscript as it is presented.

The manuscript would benefit greatly by condensing figures and reworking to provide a clearer focus.

Response: We agree that a similar serosurvey was done from retrospective samples collected in 2017 by Patil et al., but we would like to mention that in our study we also additionally conducted a prospective 30-cluster cross-sectional study in Pune city using 30-cluster sampling approach with probability-proportion-to-size (PPS) sampling, during early 2019. Our retrospective samples, collected in 2009 employing the same sampling technique, gave us the advantage of demonstrating a decadal change in seroprevalence in Pune City, which would not have been possible without a prospective study component. 

As highlighted by the reviewer, the KAP findings in the manuscript were not integrating with our manuscript, which we have thus dropped from the same.

General Concerns:

Comment 1: Having the 2009 data provides an interesting perspective regarding a potential level of immunity at which CHIKV is able to cause an epidemic (2016) but this is not developed in the manuscript. It is also unclear if the 2009 samples are similar to the 2019 samples. That is, are the 2009 samples from the community like the 2019 samples?

Response: The samples collected in 2009 were collected for the purpose of H1N1 testing, which we later tested for anti-CHIKV IgG, helping us document and understand the change in seroprevalence over a span of 10 years. The data on reported number of chikungunya cases available from Pune Municipal corporation for years 2012-2019 was studied and revealed an outbreak in 2016, which was also accurately predicted in our study by a combination of outbreak and constant model (Page 10, Lines 270-273). 

 Comment 2:  Figures 1 and 4 are nearly the same figure. It would be better to show the graph in figure 4 only.

Response: Both Figure 1 and 4 have now been modified to represent different information, Figure 1 (Page 3) shows the geo-reference map of locations where the survey was conducted in Pune City, and Figure 4 (Page 8) represents the seroprevalence of CHIKV infection in different areas as depicted by gradient of colours.

Comment 3:  Figures 3 and 5 would be more useful combined and in a table format. For Figure 5, a line graph is not appropriate as it demonstrates a linear increase from 2009 to 2019 which is not expected to be the case with an epidemic in 2016 as shown from data in Figure 6D.

Response: As suggested by the reviewer, we have now combined Figure 3 and 5 into a single table format, and shown our study findings (Page 6, Table 1).

Comment 4: It’s not clear how the KAP data integrates with the rest of the manuscript. It should either be better integrated somehow or reworked as a separate submission.

Response: We thank the reviewer for the suggestion. In the revised version, we have removed KAP data from the manuscript.

Comment 5: There is no discussion why certain wards show higher seroprevalence. For example, are these more impoverished wards, higher population density, etc? How does this translate into useful public health information going forward?

Response: We thank the reviewer for highlighting this important aspect. We reviewed our findings and found that this difference in seroprevalence could be attributed to the percentage of population living in slums, in the specific wards. We have added this vital information in the results (Page 7, Lines 228-231) and discussion (Page 11, Lines 326-335), also emphasising the impact of poor living conditions on public health. We also compared seroprevalence in slum and non-slum locations in our survey which revealed that clusters in slum had significantly higher seroprevalence than non-slum clusters. This has been added in revised manuscript.

Comment 6:  Why was an assessment of febrile illness in the past 3 months included in the manuscript when the primary objective is determining past infection (seroprevalence)? Inclusion of this detracts from the overall aim of the manuscript.

Response: As we were conducting the serosurvey prospectively on field, we wanted to utilise this opportunity to document continuous or high transmission levels, in case present, by seeking history of febrile illness in the past 3 months and conducting anti-CHIKV IgM (Page 7, Lines 220-221) .

Round 2

Reviewer 1 Report

All comments have been adressed but some new questions remained:

could you indicate the river on your map ?

In the randomised multistage cluster sampling you should indicate that somehow by chance, the sampling site are sufficiently far from each other to involve different mosquitoes population as the aedes aegypti have only a short range of dissemination by themselves. IN the discussion you outlighted the slum areas with high human density as correlated with high prevalence that is nice but what about the human population shift from work to bed that might help to the epidemic dissemination. DO you have data about ?

However it remains some typeset errors (space in double as line 79; 84; 254; line 318 double point as line 60; a space before the comma line 81 etc..).

A carrefull reading is thus still needed

As an example the population number line 79 is not correct ! there was a number missing between the comma: what is the meaning of 31,19,901 ?

In the figure 1, in the Indian Map, there were too much district in black compared to the Pune City area. Please correct. The legend need to be complemented and placed in the right place (ie under the figure) and not on the top of the page.

Line 112, I do not understand the sentence : "The households were selected by systematic sampling for the survey of individuals."

Author Response

We thank the reviewers for their critical review and valuable comments. Detailed point-by-point response to the comments is provided below, and revisions in the manuscript are provided in ‘track changes’.

Comments and Suggestions for Authors

All comments have been addressed but some new questions remained:

Comment 1: Could you indicate the river on your map ?

Response: As suggested by the reviewers, we have now shown the Mula and Mutha river in the Pune City map (Page 3, Figure 1).

Comment 2:  In the randomised multistage cluster sampling you should indicate that somehow by chance, the sampling site are sufficiently far from each other to involve different mosquitoes population as the aedes aegypti have only a short range of dissemination by themselves.

Response: During randomised multistage cluster sampling, the locations were spatially distributed to have geographic representation of the wards, as mentioned in methodology, (Page 3, Line 111-112) and we made efforts to ensure the clusters too close-by to each other were not selected for sampling, by employing a slight modification in sampling approaches.

As Aedes aegypti has a short flight range of 50-100 metres, it was ensured that the selected wards were sufficiently far from each other, so as to involve different mosquito population.

Comment 3: IN the discussion you outlighted the slum areas with high human density as correlated with high prevalence that is nice but what about the human population shift from work to bed that might help to the epidemic dissemination. DO you have data about ?

Response: We thank the reviewer for highlighting this important aspect. This is one of the limitations of our study, which we have now added in the draft (Page 8, Line 297-299). We did not enquire this aspect regarding distance of workplace from the households and mode of travel, which could have helped in understanding its importance.

Comment 4: However it remains some typeset errors (space in double as line 79; 84; 254; line 318 double point as line 60; a space before the comma line 81 etc..).A careful reading is thus still needed

Response: We thank the reviewers for pointing out the typeset errors. We have corrected them at all necessary places in the draft.

Comment 5: As an example the population number line 79 is not correct ! there was a number missing between the comma: what is the meaning of 31,19,901 ?

Response: We have corrected the decimal notation to 3,119,901. We would like to confirm that the number mentioned is correct (Page 2, Line 79).

Comment 6: In the figure 1, in the Indian Map, there were too much district in black compared to the Pune City area. Please correct. The legend need to be complemented and placed in the right place (ie under the figure) and not on the top of the page.

Response: As suggested, we have now made the necessary changes in the Figure 1 (Page 3).

Comment 7: Line 112, I do not understand the sentence : "The households were selected by systematic sampling for the survey of individuals."

Response: In our study, 30 clusters were selected from a sampling frame of 144 clusters (wards) by Population Proportional to Size (PPS) sampling. The households were listed and approximately 50-60 individuals were to be selected per cluster, so as to achieve the necessary sample size. From each age-group, a pre-determined number of samples was selected by systematic sampling, such that the samples in each age group reflected the age-wise stratification as observed in age-sex structure of population in Pune City.

We have now modified the methodology to bring more clarity regarding the sampling method (Page 3, Line 113-115).